# Novel Approach toward the Forming Process of CFRP Reinforcement with a Hot Stamped Part by Prepreg Compression Molding

**DOI:** 10.3390/ma15144743

**Published:** 2022-07-06

**Authors:** Jae-Hong Kim, Yong-Hun Jung, Francesco Lambiase, Young-Hoon Moon, Dae-Cheol Ko

**Affiliations:** 1ERC for Innovative Technology on Advanced Forming, Pusan National University, Busan 46241, Korea; kjh86@pusan.ac.kr (J.-H.K.); fmnvfmnv@naver.com (Y.-H.J.); 2Department of Industrial and Information Engineering and Economics, University of L’Aquila, 67100 L’Aquila, Italy; francesco.lambiase@univaq.it; 3School of Mechanical Engineering, Pusan National University, Busan 46241, Korea; yhmoon@pusan.ac.kr; 4Department of Nanomechatronics Engineering, Pusan National University, Busan 46241, Korea

**Keywords:** carbon fiber reinforcement plastic (CFRP), thermoforming analysis, hot stamping, prepreg compression molding (PCM), B-pillar

## Abstract

The use of carbon fiber-reinforced plastics (CFRP) is markedly increasing, particularly for the manufacturing of automotive parts, to achieve better mechanical properties and a light weight. However, it is difficult to manufacture multi-material products because of the problems due to the adhesive between CFRP and steel. The prepreg compression molding (PCM) of laminated CFRP can reduce the production time and increase the flexibility of the manufacturing process. In this study, a new manufacturing process is proposed for CFRP reinforcement on a hot stamped B-pillar using PCM. A finite element (FE) simulation of the hot stamping process is conducted to predict the dimensions of the B-pillar. The feasibility of PCM manufacturing is explored by the simulation of the thermoforming of a CFRP set on a shaped B-pillar. The temperature conditions of the CFRP and B-pillar for the PCM are determined by considering the heat transfer between the CFRP and steel. Finally, the PCM of the B-pillar consisting of steel and CFRP was performed to compare with the analytical results for verification. The evaluation of the B-pillar was conducted by the observation of the cross-section for the B-pillar and interlayer by scanning electron microscopy (SEM). As a result, a steel/CFRP B-pillar assembly could be efficiently manufactured using the PCM process without an additional adhesive process.

## 1. Introduction

Reducing the weight of automobiles using innovative materials is an attractive way to solve environmental problems and reduce fuel consumption. Carbon fiber-reinforced plastic (CFRP) is widely used in various industries, including the aerospace, machinery, sports equipment, and automobile industries. It has been shown to offer significant weight savings compared to traditional materials without a decline in mechanical properties. Numerous attempts have been made to substitute steel products with CFRP products, which have comparable mechanical properties such as stiffness and strength [1,2,3]. Conventional composites are commonly used as structural materials in industrial sectors that require a good performance over a range of temperatures and deformation rates. Components made of CFRP are commonly manufactured using an autoclave and a resin transfer molding (RTM) process [4,5]. However, these manufacturing processes require additional equipment and long cycle times to produce automotive parts. Prepreg compression molding (PCM) is an alternative process for solving the abovementioned problems [6,7].

The PCM process is required to apply CFRP to automotive parts for mass production. Several attempts have been made to use CFRP with PCM for industrial parts. Lee et al. [8] proposed a design method for manufacturing B-pillar reinforcements made of CFRP by considering the lay-up angles and the number of plies. Kim et al. [9] manufactured a B-pillar with the outer panel made of steel and the reinforcement made of CFRP using vacuum-assisted resin transfer molding (VA-RTM). Frantz et al. [10] manufactured a hybrid product by forming a prepreg on existing steel components. However, the aforementioned studies used additional adhesive processes to add each part.

Various researchers also studied the feasibility of the manufacturing process using CFRP and steel parts Yanagimoto et al. [11] evaluated the formability of CFRP overlapped with a metal sheet by simple forming, such as embossing, hat bending, and V-bending. Heggemann et al. [12] used a combination of high-strength steel and CFRP prepregs in a special hybrid material/fiber metal laminate (FML) in the deep drawing process. Wollmann et al. [13] investigated the formability of the CFRP prepreg and steel sheet using the analytical prediction method. Lee et al. [14] performed deep drawing tests for hybrid materials consisting of a CFRP prepreg and steel sheets to evaluate forming behaviors. However, in these studies, the application of ultrahigh-strength steel used in the hot stamping process has not been performed to adapt CFRP to complex shapes, such as automotive parts.

This study proposes a new manufacturing process using a PCM on a hot stamped B-pillar without an additional adhesive process for bonding steel and CFRP. The process was organized into two stages: hot stamping of the B-pillar and PCM process of the CFRP reinforcement of the manufactured product. First, the process conditions to manufacture B-pillars without defects were determined using the finite element (FE) simulation of hot stamping. The feasibility of PCM manufacturing is explored by the simulation of the thermoforming of a CFRP set on a shaped B-pillar. The hot stamped B-pillar, which was manufactured in the first stage, was used as the lower die in this stage. Additionally, the heat transfer between the CFRP and steel was considered in the FE simulation to verify the feasibility of this process. Next, an experiment on the PCM process was performed to manufacture the B-pillar assembly under the same conditions as the FE simulation, and the results were compared with the analysis results. For verification, various observations using scanning electron microscopy (SEM) were performed to investigate the forming and interlayer behavior between the steel and CFRP.

## 2. Materials and Method

### 2.1. Mechanical Preperties of the Steel Sheet and CFRP

In this study, boron steel (22MnB5) was used to manufacture the B-pillar by the hot stamping process. The mechanical properties of boron steel were obtained from a hot tensile test considering the forming temperature [15]. The Cowper–Symonds model was adopted as a constitutive equation to describe the material behavior of boron steel by considering the temperature and strain rate, and it can be expressed as follows [16];
(1)$$\sigma_{dyn}=\sigma_{stat}\cdot {\left[1+\left(\frac{\dot{\varepsilon }}{C}\right)\right]}^{\frac{1}{p}},$$
where $\sigma_{dyn}$ is the dynamic stress; $\sigma_{stat}$ is the quasi-static stress at a strain rate of 0.1/s; $\dot{\varepsilon }$ is the strain rate; and *C* and *p* are the material constants related to the strain rate and temperature, respectively.

The thermomechanical properties of boron steel are applied to perform the FE simulation of hot stamping, as presented in Table 1. Therefore, the FE simulation of hot stamping could describe the deformation behaviors of boron steel and the heat transfer between the tool and steel sheet.

The CFRP used in this study was a commercial twill weave prepreg provided by SK Chemicals. The carbon fiber (MRC PYROFILTM TR30S 3 K) used in the prepreg was manufactured by Mitsubishi Rayon. The resin was a polyester-based thermoplastic polyurethane with a glass transition temperature (T_g_) of 110 °C. The thickness of one ply was 0.3 mm, and the volume fraction and density of the carbon fiber were evaluated as 39.64% and 1.52 g/cm^2^, respectively. The mechanical properties of the CFRP are listed in Table 2.

### 2.2. Forming Process of CFRP Reinforcement on a Hot Stamped Part

Figure 1 shows a schematic diagram for the forming process of CFRP reinforcement on a hot stamped part. In the first stage, the hot stamping process was designed by FE simulation to manufacture the B-pillar of boron steel. In this study, the formability of the hot stamped B-pillar was evaluated by the forming limit diagram (FLD) of boron steel at high temperatures. The spring-back after the hot stamping process was also measured to obtain a high dimensional accuracy of the B-pillar.

In the second stage, the PCM process was designed by FE simulation to manufacture the reinforcement of CFRP. In this study, the hot stamped part was used as the lower die, and the process conditions for PCM were determined to prevent various defects, such as wrinkles, delamination, and insufficient forming. The temperature histories of CFRP were important factors to manufacture the CFRP part, and various tool temperatures were therefore considered in the FE simulation.

Figure 2 shows the procedure for the process design of the steel/CFRP B-pillar assembly. The advantages of this process are the light weight of the automotive parts, the reduced parts and forming processes, and the removal of the bonding process. In addition, it can be applied to various existing parts, which require reinforcement.

## 3. Process Design of the B-Pillar Assembly by FE Simulation

### 3.1. FE Simulation of the Hot Stamping Process

In this study, a hot stamping process was designed considering the actual process conditions. A high dimensional accuracy was required because a hot stamped B-pillar was used in the tool for the PCM process to manufacture the CFRP reinforcement. Therefore, the spring-back behavior of the B-pillar had to be precisely predicted by the FE simulation. In addition, formability was investigated according to FLD.

The FE simulation of the hot stamping process was performed using the commercial software JStamp/NV, and the FE model is shown in Figure 3. The FE model consists of an upper and lower die and an initial blank, as the size of the initial steel blank was W240 mm × L510 mm × t1.4 mm. The specimen was a shell element with a uniform size of 3.0 mm × 3.0 mm, and the tool was assumed to be a rigid body.

An FE simulation was performed for the entire hot stamping process, including the heating, transferring, forming, and quenching stages. In the heating stage, the initial blank was heated to 950 °C for 5 min. Subsequently, the heated blank was transferred to the die within 10 s, and the blank was formed by a forming punch at 20 mm/s for 2 s. Finally, the formed blank was quickly quenched via heat transfer between the tool and the blank for 10 s.

Figure 4 shows the formability evaluation of the B-pillar by comparison with the FLDs under different temperatures [20]. The temperature of the boron steel decreased during the forming stage owing to the contact between the tool and blank, which could lead to problems, such as cracking and wrinkling. Therefore, it is essential to evaluate the results of the forming analysis by the FLDs of each material; as shown in the graph, there were no problems during the hot stamping.

In the hot stamping process, the dimensional accuracy of the B-pillar significantly affects the forming process for the CFRP reinforcement. In this study, three cross-sections of the predicted B-pillar were selected to measure the spring-back for comparison with the target product. During the hot stamping process, forming and cooling occur simultaneously. Thus, a phase transformation occurs from the austenite phase to the martensite, which could have an effect on the dimensions of the B-pillar [21]. Therefore, the spring-back, after the quenching process, was evaluated by simulation.

Figure 5 shows the method and location for measuring the value of the spring-back on the shaped B-pillar by hot stamping simulation. The benchmark of NUMISHEET‘93 was used as the measurement method. Four different points were chosen to measure the angles for each cross-section. The measured angles were compared to those of the target product at the same position.

Figure 6 shows a comparison of the spring-back between the FE simulation and the target angle. In the cross-section at A, the angles $\theta_{1}$, $\theta_{2}$, $\theta_{3}$, and $\theta_{4}$ from the FE model were observed to be 142°, 97°, 92°, and 103°, respectively. The angles of the original model at the same position were measured as 138°, 93°, 91°, and 100°, respectively. The comparison showed that the B-pillar exhibited high dimensional accuracy in the case of sections (B) and (C). Therefore, the designed B-pillar can be used as a tool for the PCM process.

### 3.2. FE Simulation of the PCM Process

Generally, the thermoplastic CFRP prepreg hardens immediately when the temperature of the CFRP decreases below T_g_ owing to the heat transfer between the tool and laminate. Therefore, the temperatures of the material and tool during forming are the primary process parameters. To determine the thermoforming conditions, an FE simulation of the PCM was conducted using the commercial FE software (PAM-FORM 2020.0). First, the initial heating temperature of the laminate was determined to be 200 °C, which is the maximum heating temperature of the polyurethane and prevents the heat loss of the laminate owing to the atmospheric air. The laminate comprised four plies with CFRP prepregs modeled with the dimensions of 88 mm × 130 mm in [0°]_4_. The dimensions of the B-pillar obtained from the analytical results of the hot stamping were applied to the thermoforming analysis of CFRP as the lower die.

The effect of tool temperature should be investigated to determine the forming condition of the PCM. Generally, when the forming temperature of the tool is low, the heating and cooling times are reduced. Thus, various temperature conditions, namely, 90, 110, 130, and 150 °C, were selected as the heating temperatures of the punch and lower die. Non-impregnation caused by forming below T_g_ was predicted by investigating the temperature history of the laminate. The friction coefficient of the CFRP laminate was applied as a reference to consider the interface behavior [19]. The heat transfer coefficients of steel and CFRP were used to predict the heat loss of the product that affected the formability. The analysis of forming was conducted using the following process: an initial stage, a gravity stage, and a forming stage using heated tools. The FE model is shown in Figure 7.

Figure 8 shows the temperature histories of the laminate during the thermoforming analysis. The tool temperature influenced the formability of the CFRP product. In the case of a tool temperature of 90 °C, it was confirmed that the temperature of the laminate rapidly decreased to below T_g_ owing to the heat transfer to the tool and laminate during the forming process. However, when the tool temperature was higher than 130 °C, the minimum temperature of the laminate was maintained above T_g_ until the forming stage was completed. The CFRP represents a low formability below T_g_ but improves rapidly above T_g_ because of matrix softening and the loss of fiber–matrix adhesion. It was important to estimate the temperature of the CFRP laminate because the analytical results of the thermoforming did not directly indicate the non-impregnation. Therefore, 130 °C was selected as the tool temperature for an optimum forming process.

## 4. Results of Experimental Verification

### 4.1. Hot Stamping Process for Manufacturing the B-Pillar

In this study, an experiment for hot stamping was performed to validate the proposed process for manufacturing a steel/CFRP B-pillar assembly. The equipment for the experiment consisted of a furnace for heating, a tool with upper and lower dies for hot stamping, and a 1000-t hydraulic press, as shown in Figure 9. Additionally, a water-cooling system was used as an embedded channel for the quenching stage. The initial blank used in the experiment was 22MnB5 boron steel with dimensions of W240 mm × L510 mm × t1.4 mm, which was the same shape as the blank used in the analysis of the forming process. The blank was heated in an electric furnace and transferred to a die. After the transfer stage, a blank was formed using the upper punch and quenched. All the conditions in the experiment were the same as those in the FE simulation. According to the procedure, the lab-scale B-pillar was manufactured with no defects, as shown in Figure 10. Figure 11 shows a comparison of the spring-back between the FE simulation and the experiment. The B-pillar was applied as the lower die in the PCM process.

### 4.2. PCM Process for Manufacturing the CFRP Reinforcement on the B-Pillar

To verify the proposed process, the PCM process was used to manufacture the steel/CFRP B-pillar assembly under the same conditions as the thermal analysis. The apparatus used in this experiment is shown in Figure 12. The tools comprised an upper punch with an inserted cartridge heater and a lower jig to hold the heated B-pillar on a 200-t servo press. Two different heating chambers were operated to heat the hot stamped B-pillar used as the lower die and the CFRP laminate with the four-plies prepreg. The lay-up and dimensions of the CFRP laminate were the same as those used during the thermoforming analysis. The CFRP products were manufactured at tool temperatures of 90, 110, and 130 °C.

The manufacturing process of the steel/CFRP B-pillar assembly was as follows. First, a liquid release agent (Chemlease 15 Sealer EZ) was spread on the upper punch surfaces to prevent the CFRP laminate from sticking to the upper punch. Next, the laminate was heated to 200 °C in the chamber, and the hot stamped B-pillar was heated to the target temperature. Subsequently, the laminate was transferred to the heated B-pillar before the B-pillar was moved to the lower jig. When the B-pillar was positioned at the lower jig, the upper punch was moved to perform the forming process. After the tools were cooled to the lower temperature of T_g_ by air quenching, the product was ejected from the tools, as shown in Figure 13.

### 4.3. Evaluation of the Experimental Result of the Steel/CFRP B-Pillar Assembly

Figure 14 shows the experimental results for the B-pillar by the PCM process. The results were similar to those obtained from the FE simulation; therefore, the effectiveness of the forming analysis was confirmed. In the case when the tool temperature was 90 °C, wrinkles were observed on the CFRP surface and interface delamination on the side of B-pillar assembly because of the temperature being lower than T_g_. In the case of the tool temperature of 110 °C, interface delamination between the steel and CFRP showed gradual improvement, although the wrinkles on the CFRP surface remained. When the tool temperature was 130 °C, which was determined as an appropriate condition, there were no defects, such as wrinkles or delamination. Essentially, this condition offers good formability compared with the other cases.

Cross-sectional observations were performed to figure out the forming and adhesive behaviors, as shown in Figure 15. Various sections were selected to observe the interlayer delamination in the upper and lower sections and the formability and adhesive behavior in the middle section. The selected sections were cut by water jet cutting and observed depending on each forming condition. In the case of the temperature of 90 °C, insufficient forming of the CFRP was observed on all the cross-sections. In addition, the CFRP reinforcement fell off the B-pillar during the water jet cutting. In the case of the temperature of 110 °C, the CFRP reinforcement exhibited better quality than the previous case but was still not satisfactory, with inadequate forming on the corner and bottom of the B-pillar. In contrast, in the case of the temperature of 130 °C, the major problems of the previous two results were resolved in the entire B-pillar.

Hardness measurements by an automatic hardness testing system (Mitutoyo HM-200) were performed with an indentation load of 500 g for a dwell time of 10 s to observe the change in the hardness of the B-pillar during additional heating in the PCM process. Figure 16 shows a comparison of the measured hardness between the hot stamp and PCM with a tool temperature of 130 °C. The specimens were prepared by wire cutting and were checked at four different points. There was no difference in hardness between the hot stamped and test cases in the comparison results. The interlayer between the steel and CFRP was observed to check for the adhesive behavior. Figure 17 shows the interlayer conditions at 130 °C by SEM. The observed interlayer between the steel and CFRP represented a boundary line similar to that with the clad metal [22]. The soundness of the bonded interface between the steel and CFRP is intact even though the adhesive is not used for B-pillar assembly after the PCM process. Therefore, the proposed design method is feasible for manufacturing steel/CFRP B-pillars using PCM without additional adhesive processes. Additionally, the presented process has advantages in terms of manufacturing cost and time compared with the existing process because the hot stamped part takes over the role of the lower tool, and this process could be easily applied to various manufactured parts.

## 5. Conclusions

In this study, a steel/CFRP B-pillar assembly was manufactured using the PCM process without an additional adhesive process. The process design method was proposed based on the FE simulation, and the results are as follows. The B-pillar was manufactured using a hot stamping process, wherein the conditions were determined using FE simulation. The results of the formability evaluation indicated that there were no problems during hot stamping. The B-pillar has high dimensional accuracy, as shown in the comparison results of the spring-back between the FE simulation and target model. Therefore, the designed B-pillar can be used as a tool for the PCM process. The thermoplastic CFRP prepreg hardened immediately when the temperature of the CFRP decreased below T_g_ owing to the heat transfer between the tool and laminate. Therefore, the temperatures of the tools were determined via FE simulation using the predicted dimensions of the B-pillar. In the case of the tool temperature of 130 °C, the minimum temperature of the laminate was maintained above T_g_ until the forming stage was completed. The experiments for hot stamping and PCM were performed to validate the proposed method for CFRP formation on a hot stamped B-pillar. The manufactured steel/CFRP B-pillar assembly with forming conditions obtained from the FE simulation presented no defects, such as wrinkles, non-impregnation, and delamination. Therefore, the method proposed in this study will help in designing the forming process for steel and CFRP for various automotive parts.

The future directions of this study involve further investigation to improve the accuracy of the prediction by considering the thermomechanical properties of various 3D woven carbon fiber-epoxy composites and steel sheets. Additionally, the method suggested in this study will be applied to the design of the manufacturing process of automotive parts considering recyclability. Additionally, the crash or structural performance of steel/CFRP assembly will be evaluated by analytical and experimental studies.

## Figures and Tables

**Figure 1 materials-15-04743-f001:**
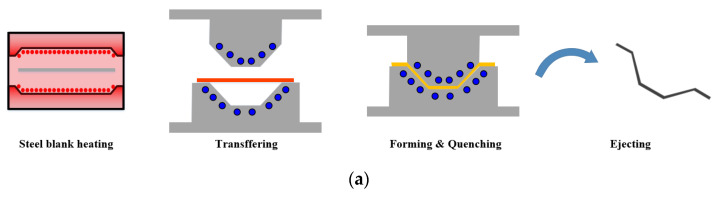
Schematic diagram for the manufacturing of the B-pillar assembly. (**a**) Hot stamping process for the manufacturing of the B-pillar using boron steel. (**b**) PCM process for the manufacturing of the reinforcement using CFRP.

**Figure 2 materials-15-04743-f002:**
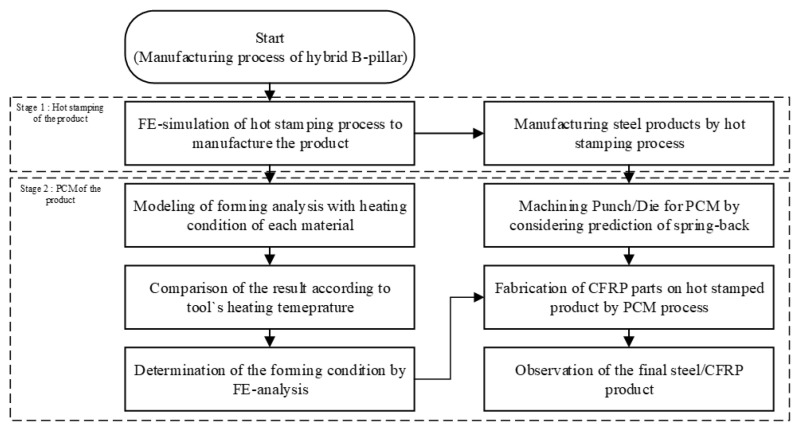
Procedure for the process design of the steel/CFRP B-pillar assembly.

**Figure 3 materials-15-04743-f003:**
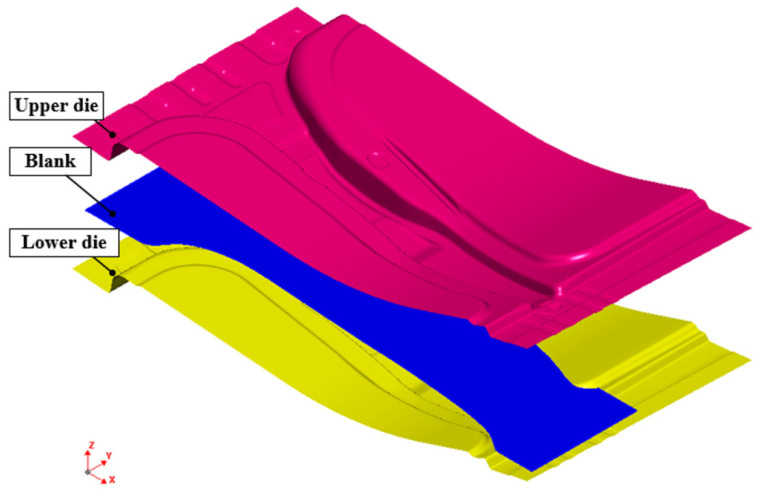
FE model for the hot stamping of the B-pillar.

**Figure 4 materials-15-04743-f004:**
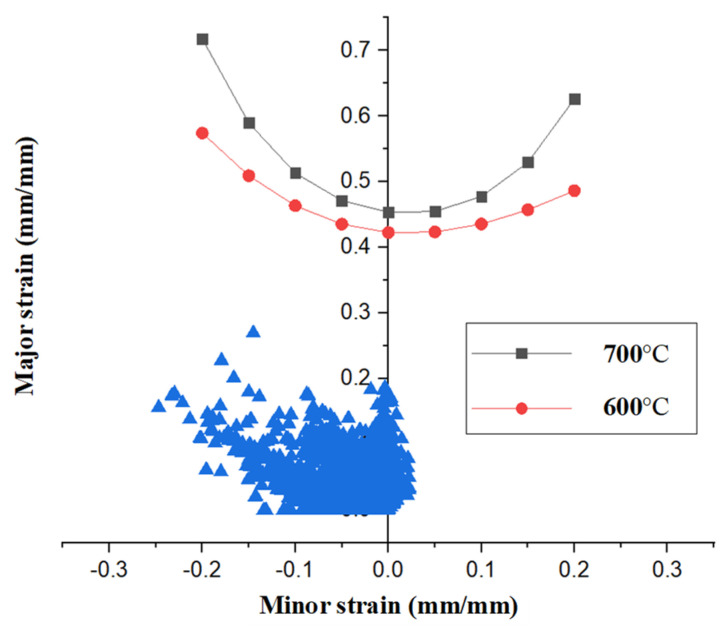
Formability evaluation with FLDs of 22MnB5 for high temperatures.

**Figure 5 materials-15-04743-f005:**
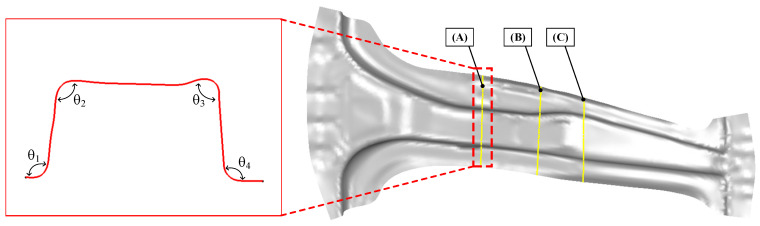
Measuring sections of the B-pillar for the evaluation of the spring-back behavior.

**Figure 6 materials-15-04743-f006:**
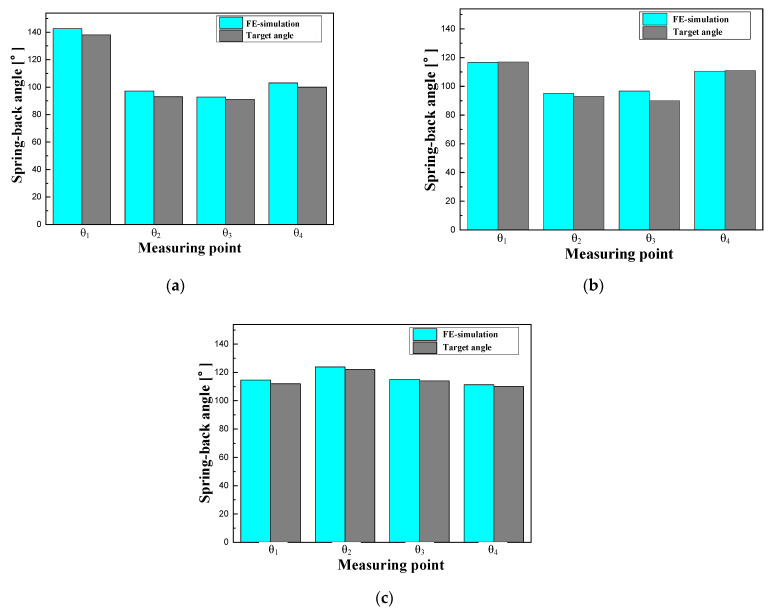
Comparison of the spring-back between the analytical results and target angle. (**a**) Cross-section at A. (**b**) Cross-section at B. (**c**) Cross-section at C.

**Figure 7 materials-15-04743-f007:**
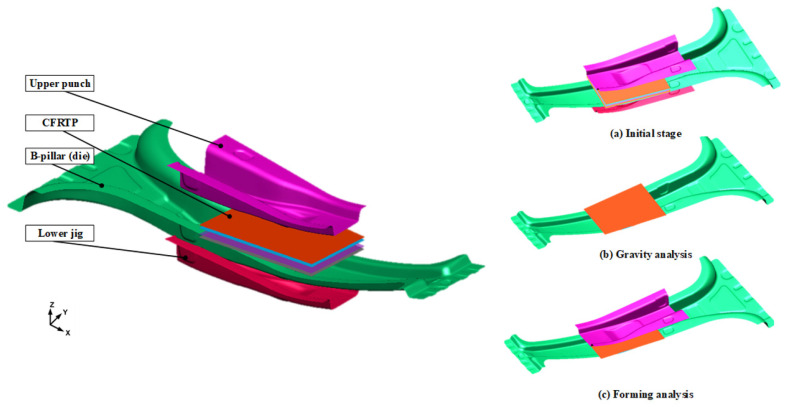
FE model and processes of thermoforming analysis of CFRP reinforcement on the B-pillar.

**Figure 8 materials-15-04743-f008:**
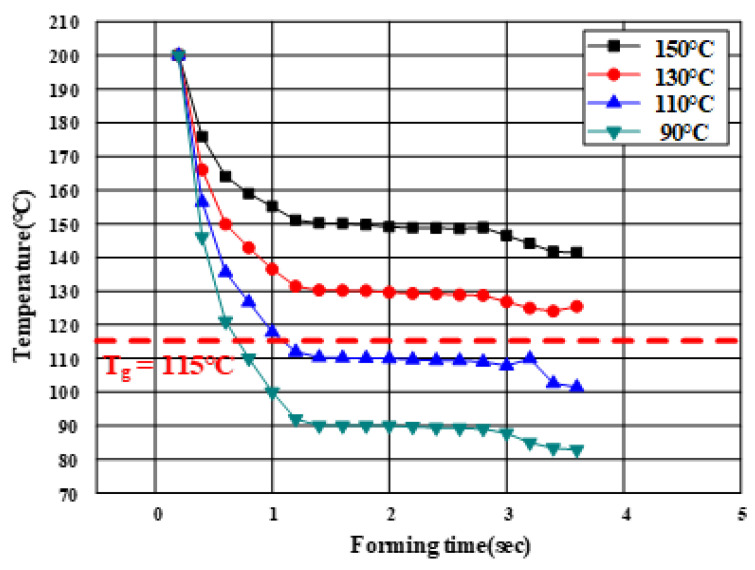
Temperature histories of laminate during the PCM process.

**Figure 9 materials-15-04743-f009:**
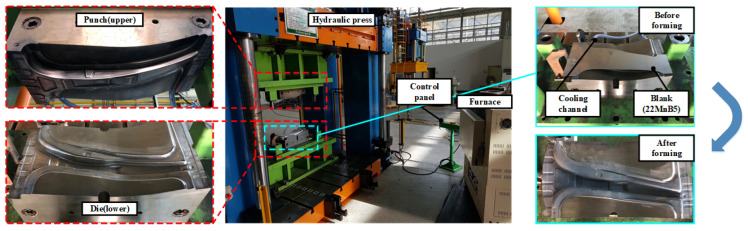
Experimental equipment for the hot stamping of the B-pillar.

**Figure 10 materials-15-04743-f010:**
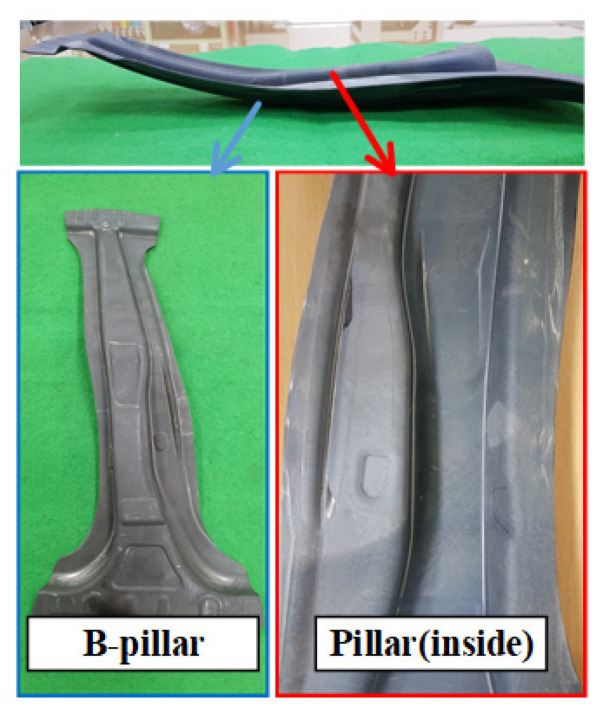
Manufactured B-pillar by the hot stamping process.

**Figure 11 materials-15-04743-f011:**
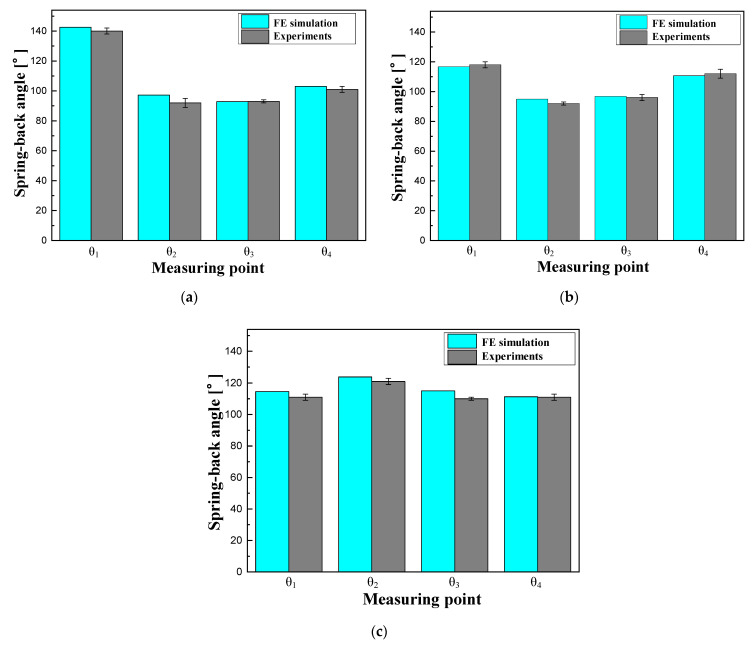
Comparison of the spring-back between the analytical results and experimental results. (**a**) Cross-section at A. (**b**) Cross-section at B. (**c**) Cross-section at C.

**Figure 12 materials-15-04743-f012:**
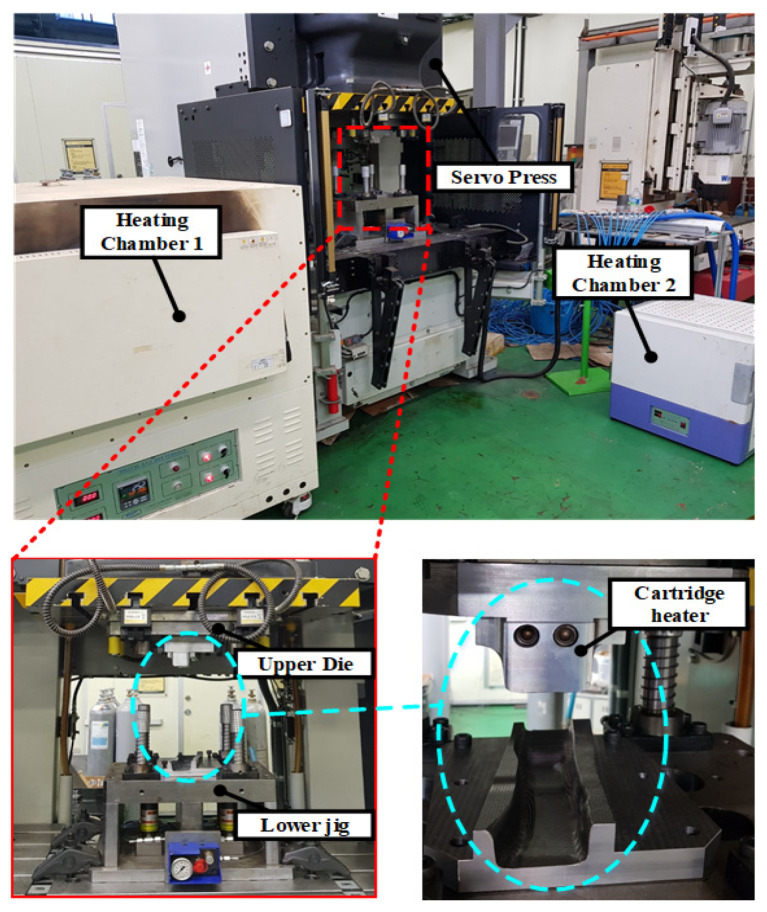
The experimental apparatus for the PCM process.

**Figure 13 materials-15-04743-f013:**
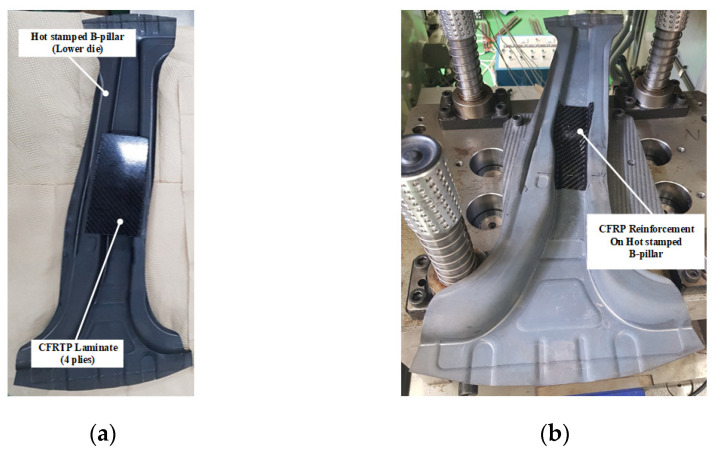
CFRP reinforcement on the hot stamped B-pillar by PCM. (**a**) Before forming. (**b**) After forming.

**Figure 14 materials-15-04743-f014:**
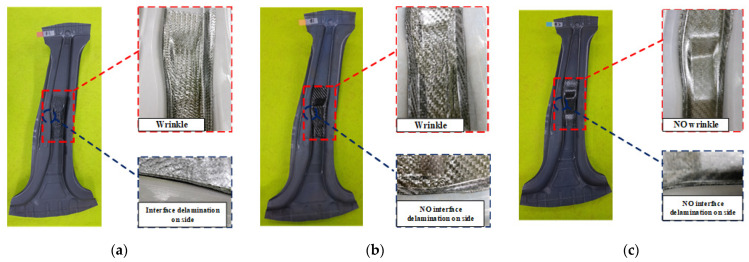
Observation of defects at different tool temperatures. (**a**) 90 °C. (**b**) 110 °C. (**c**) 130 °C.

**Figure 15 materials-15-04743-f015:**
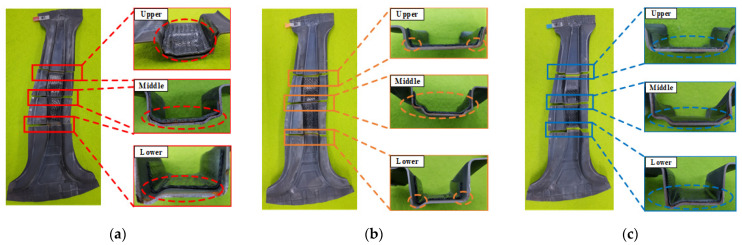
The cross-sectional observation according to various tool temperatures. (**a**) 90 °C. (**b**) 110 °C. (**c**) 130 °C.

**Figure 16 materials-15-04743-f016:**
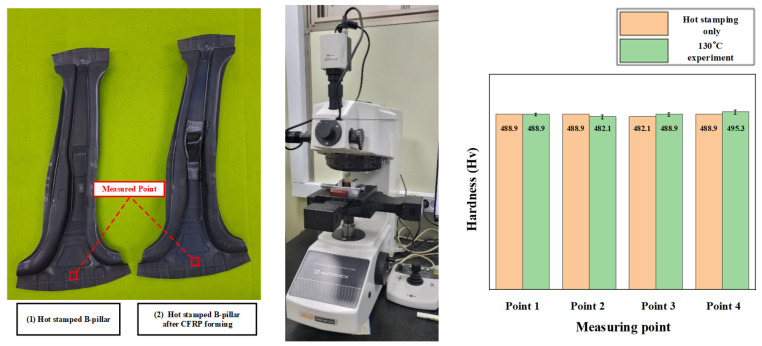
Comparison of the hardness between the hot stamped and PCM-tested B-pillar.

**Figure 17 materials-15-04743-f017:**
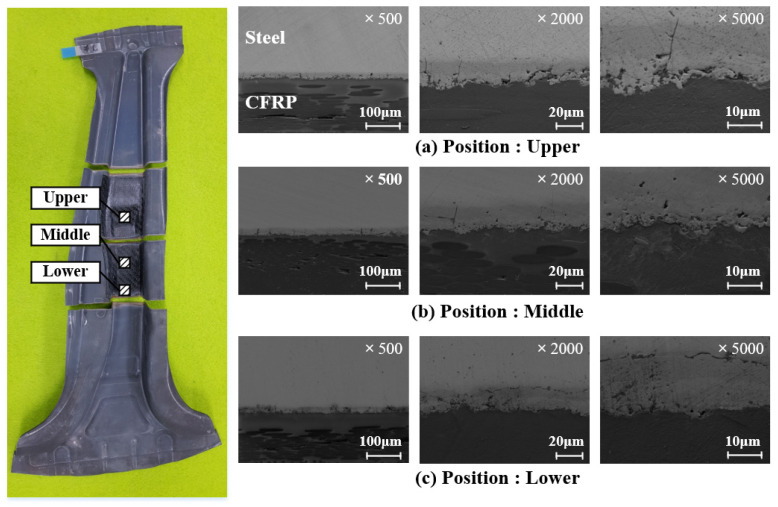
Observation results of the interlayer behavior by SEM.

**Table 1 materials-15-04743-t001:** Material constants of boron steel in the FE simulation [17,18].

Conditions	Values
Young’s modulus (GPa)	As a function of pressure
Poisson’s ratio	0.3
Thermal expansion (1/K)	$1.44\times 10^{-5}$
Heat conductivity (W/m∙K)	32
Convective heat transfer coefficient (W/m²∙K)	20
Interfacial heat transfer coefficient (W/m²∙K)	As a function of pressure
Friction coefficient (μ)	0.4

**Table 2 materials-15-04743-t002:** Mechanical properties of CFRP [19].

Mechanical Properties	Values
Elastic modulus in fiber direction (E_11_) (GPa)	40.35
Elastic modulus in transverse direction (E_22_) (GPa)	40.35
Shear modulus in 1–2 plane (G_12_) (GPa)	9.51
Shear modulus in 2–3 plane (G_23_) (GPa)	0.30
Shear modulus in 1–3 plane (G_13_) (GPa)	0.30
$\mathrm{Poisson}’s\;\mathrm{ratio}\;(\upsilon_{12}$) (GPa)	0.13

## Data Availability

The data presented in this study are available on request from the corresponding author and on reasonable request from the first author.

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
