# Peer review of "Novel Approach toward the Forming Process of CFRP Reinforcement with a Hot Stamped Part by Prepreg Compression Molding"

_materials, 2022, doi:10.3390/ma15144743_

Round 1
Reviewer 1 Report
The paper seeks to introduce an approach; Novel Approach toward the Forming Process of CFRP Reinforcement with Hot Stamped Part by Prepreg Compression Molding. However, the authors should consider to improve upon the quality to further highlight and emphasis.
1) Based on the understanding of what should be included in the abstract;
Abstract of any scientific articles should address three main elements;
i. Description of the problem
ii. Description of the research findings
iii. Significance of the study
In view of these, introduce one or two lines to highlights the significance of your study at the end of the abstract.
2) Each figure especially should be accompanied with the unit of the representative of the x and y axes. Figure 3 has no unit attached, thus, making it unitless. Consider adding units.
3) The introduction needs to be improved by relating to the mechanics of the studied materials and their mechanical characteristics. The references to be included are: 10.1007/s10853-022-06994-3, 10.1177/0021998318790093, 10.1016/j.polymertesting.2017.09.009.
4) All the units in table 2 should be moved to the value section inside the table in parenthesis instead of adding to each value in the table
5) The footers (magnification bar) in the SEM images are not visible. Manually indicate each inside the figures
6) In section 4.2, you mention that CFRP laminate is prevented from sticking to the surface of the of the upper punch. What is the name of the liquid release agent and what will happen if the laminate stick to the surface of the upper punch?
7) What characterization technique did you applied in confirming the manufactured products conforming to the simulated results in section 4.3
8) In section 4.3, paragraph 3, you made mention of a hardness test which was carried out but failed to mention the name of the device used and its schematic
Author Response
Thank you for your considerate review about the manuscript.
Please see the attachment.

Reviewer 2 Report
The manuscript entitled, i.e., Novel approach toward the forming process of CFRP reinforcement with hot-stamped part by prepreg compression molding, has been reviewed. There are several problems with the manuscript in its present form. Therefore, it cannot be recommended for publication in its present form.
Just a suggestion, avoid using the word Novel in the title. Also, as the approach to forming process is not new, it is well-established industry practice.
The quality of the manuscript is not very good. There is too much to be improved in the manuscript before considering it for publication. Some improvements are strongly needed to make this.
The introduction must be strengthened by suggesting that some authors have done a comparative study on interlaminar properties of l-shaped two-dimensional (2D) and three-dimensional (3D) woven composites or the failure mechanism of composite stiffener components reinforced with 3D woven fabrics. They have been doing this experimentally to verify the properties. Thus I appreciate authors using FE analysis rather than a time-consuming and tiring experimental process.
Please follow the IMRAD format.
Add a section about materials and experiments.
What does the author think this FE model would be useful to improve?
According to the authors, what will be the predicted/estimated production efficiency using this model? By the way, what is the efficiency of the proposed model?
I could not find any comparison of the proposed model with the existing technology. Can the author propose a comparison table or schematics to attract readers?
Usually, validation of the model is added to the manuscript. What is the validation of this model?
Add statistical error in Figures 5 and 14.
What are future recommendations? The authors might do a comprehensive study on the mechanical properties of different 3D woven carbon fiber-epoxy composites in the future, or there might be enduring prepregs containing imine-containing epoxy vitrimer with versatile recyclability and fine their application is fully recyclable carbon fiber reinforced composites.
The references cited in the manuscript are somehow old. I could not find any manuscript from the last two years in the reference section.
Author Response

(The authors gave the same response as above.)

Reviewer 3 Report
The authors submitted an interesting paper on the CFRP reinforcement forming process by using preformed, hot-stamped parts. Please find some comments to improve the quality of the paper:
1) Introduction - Lines 57 - 77 - this should be moved to the materials and methods section or experimental verification. It does not fit the introduction. Please extend the introduction significantly. The twelve references found in that section do not reflect the state of the art at a satisfactory level. Subsequently, the knowledge gap and the motivation of the paper should be clearly exposed.
2) Materials and methods section should appear in this paper. It should have all the necessary information that enables the reproduction of the work presented.
3) FE Simulation - do the experimental results as high-temperature tensile curves were used to validate the accuracy of the model used?
4) FE Simulation - authors have assumed that "the initial blank was heated to 950 °C for 5 min. Subsequently, the heated blank was transferred to the die within 10 s and the blank was formed by a forming punch at 20mm/s for 2 s"
- does the FE simulation includes the temperature drop that occurred during the transfer found during the real forming?
5) FE Simulation - Figure 3 - how the FLD has been obtained?
6) FE Simulation - Lines 127 - 132 - how the angles of the formed part were measured? What was the accuracy of the measurement?
7) Lines 168 - 177 - please discuss the dependence of the tool temperature on CFRP formability in relation to its microstructural behavior.
8) Experimental verification - Lines 193-194 - "All the conditions in the experiment were the same as those in the FE simulation." - how the temperature was monitored during the hot stamping process? Please provide the comparison of the post-forming quenching curve with this obtained from the real forming experiment.
9) Lines 209 - 210 - "The CFRP products were manufactured at tool temperatures of 90, 110, and 130 °C" - how the temperature of the tool was ensured/maintained during the forming?
10) Line 213 - "the laminate was heated to 200 °C " - again, hot the temperature of the laminate was monitored?
11) Line 223 225 - "Figure 12 shows the experimental results for the B-pillar by the PCM process. The results were similar to those obtained from the FE simulation; therefore, the effectiveness of the forming analysis was confirmed." - on what basis do the authors claim, that experimental results were similar to those from FE?
12) Figure 14 - the basic information about hardness measurement is missing (method, force, dwelling time etc.). The confidence intervals are missing.
13) Figure 15 - the contrast of the SEM images should be increased to clearly show the connection area. A discussion on the joint connection between steel and laminate should be provided. The effect of the steel component surface on the connection efficiency should be discussed as well as microstructural features affecting the formability and quality of the laminate-metal joint area.
The paper has some publication potential, however, at this stage, it is chaotic. Some important details are missing. The scientific discussion is not provided at a satisfactory level. It is more a technical report rather than a scientific paper. The reviewer will reconsider this paper after major revision.
Author Response

(The authors gave the same response as above.)

Round 2
Reviewer 2 Report
it is considerable.
Author Response
Thank you for your considerate review about the manuscript.
Unfortunately, the references by your recommendation were excluded due to the opinion of other reviewers that they had no relevance to this paper.
Again, thank you for your kind comments for my manuscript.
P.S. This manuscript has been English proofread by a professional organization and checked with some typos and once again English proofread throughout the manuscript.
Reviewer 3 Report
All my comments were addressed thus the paper could be considered for publication by the Editor/Special Issue Editor.
I would recommend enhancing the paper's quality by comparing the presented methodology with the existing forming methods in order to highlight its advantages. A short discussion would be beneficial for the paper for sure.
However, I do not recommend citing the study not related to the submitted manuscript. Fair reviewers should not force the authors to cite their papers. Please consider the removal of the references provided by the other reviewer.
